# A Highly Sensitive and Flexible Capacitive Pressure Sensor Based on Alignment Airgap Dielectric

**DOI:** 10.3390/s22197390

**Published:** 2022-09-28

**Authors:** Soo-Wan Kim, Geum-Yoon Oh, Kang-In Lee, Young-Jin Yang, Jeong-Beom Ko, Young-Woo Kim, Young-Sun Hong

**Affiliations:** 1Sustainable Technology and Wellness R&D Group, Korea Institute of Industrial Technology (KITECH), Jeju 63243, Korea; 2Institute of Advanced Technology Development, Hyundai Motor Co., Seongnam 13529, Korea

**Keywords:** capacitive pressure sensor, alignment airgap dielectric, flexible sensor, high sensitivity

## Abstract

Flexible capacitive pressure sensors with a simple structure and low power consumption are attracting attention, owing to their wide range of applications in wearable electronic devices. However, it is difficult to manufacture pressure sensors with high sensitivity, wide detection range, and low detection limits. We developed a highly sensitive and flexible capacitive pressure sensor based on the porous Ecoflex, which has an aligned airgap structure and can be manufactured by simply using a mold and a micro-needle. The existence of precisely aligned airgap structures significantly improved the sensor sensitivity compared to other dielectric structures without airgaps. The proposed capacitive pressure sensor with an alignment airgap structure supports a wide range of working pressures (20–100 kPa), quick response time (≈100 ms), high operational stability, and low-pressure detection limit (20 Pa). Moreover, we also studied the application of pulse wave monitoring in wearable sensors, exhibiting excellent performance in wearable devices that detect pulse waves before and after exercise. The proposed pressure sensor is applicable in electronic skin and wearable medical assistive devices owing to its excellent functional features.

## 1. Introduction

Pressure sensors are actively used in advanced applications in robots [1,2,3], human-machine interaction systems [4,5], healthcare monitors [6,7,8], and energy harvesting [9,10,11]. Various pressure sensors based on ceramics and polymers have been proposed in recent years [12,13,14]. In particular, a variety of polymer-based flexible sensor devices for collecting biometric information have been studied for their applicability in human healthcare monitoring [15,16], wearable devices [17,18], and medical diagnosis tools [19,20]. These applications are related to interactions between users and devices. Therefore, flexible pressure sensors should have a high sensitivity in a low-pressure range to detect blood pressure or light touches (0~10 kPa) and a wide pressure measurement range to include the entire tactile pressure range (0–100 kPa) [21].

Pressure sensors generate electric signals when they detect external pressure. Their performance is evaluated by major parameters, such as sensitivity, detection limit, operating pressure range, response time, and linearity. Their operation is based on a signal conversion mechanism [22,23], and they are generally classified into three types: piezo-resistant [24,25], capacitive [26,27,28], and piezoelectric [29]. Among these types, capacitive pressure sensors are the most popular owing to their quick response, high reversibility, temperature insensitivity, simple structure, ease of manufacture, and low power consumption. In general, a capacitive pressure sensor is composed of two parallel electrodes and a dielectric layer aligned between them. When pressure is applied to a capacitive pressure sensor, the dielectric thickness decreases, causing a capacitance change in the sensor. However, capacitive pressure sensors typically exhibit low sensitivities due to the relatively small changes in the capacitance of parallel plates[20]. Various methods to improve the performance of these pressure sensors have been studied. The most common method for improving performance is by engineering microstructures on the elastomer surface. Microstructure patterns of the dielectric layer, such as convex [30], pyramids [31], pillars [32], and waves [33], are effective for calculating the high sensitivity of sensors. In this case, higher sensitivity can be obtained, but if the pressure applied to the sensor exceeds a certain range, the sensitivity drops significantly, and the working pressure is limited to less than 10 kPa. Moreover, this method involves an expensive, time-consuming, and complex manufacturing process for preparing microstructure silicon molds.

As an alternative, a porous structure inside the dielectric layer has been suggested. As a result, various methods, such as the particle-template method [34], chemical foaming method [35], and oil painting-template method, have been studied. Among them, the particle-template method emerged with a high potential for providing a highly sensitive sensor in addition to ease of production at a low cost. Particle-template method, easily soluble particles, such as polydimethylsiloxane (PDMS) [36] or Ecoflex [37], are added to the elastomer. The effective dielectric constant can be improved by improving the dielectric constant of a porous dielectric substance. Consequently, a new strategy to improve sensitivity through a structural change of the dielectric layer of the capacitive pressure sensor is suggested.

However, it is practically impossible for the conventional manufacturing method to evenly distribute airgaps inside the dielectric layer as irregular airgaps are generated when the dielectric layer is compressed in uniform distribution, affecting the performance of sensors. Hence, the main aim of this study is to propose a structural design that can align the airgaps in the dielectric to improve the sensor’s sensitivity, linearity, and detection range.

In this study, we propose a new flexible capacitive pressure sensor that uses the precise alignment structure of airgaps inside the dielectric layer. The fabricated sensor undergoes a simple method for producing a dielectric layer. Furthermore, the performance of the sensor was improved through the alignment of airgaps of a pure elastomer dielectric that does not contain any functional material. Various geometric structures of capacitive pressure sensors of the airgap alignment method were designed, and the detection mechanism of the devices was systematically analyzed. In addition, the responses of the pressure sensor for different sizes and spacings of internal airgaps were experimentally characterized. Lastly, the developed sensor was tested for human pulse wave monitoring.

## 2. Experimental Details

The mold and micro-needle for the generation of alignment airgaps were designed as shown in Figure 1a. The mold for manufacturing the dielectric layer was fabricated with an internal size of 20 mm × 20 mm × 3 mm. It was fabricated with an aluminum alloy (A6061) with airgaps of 300 μm, 400 μm, and 500 μm, and spacings of 1 mm, 1.5 mm, and 2 mm to form alignment airgaps. The micro-needle was fabricated with a length of 70 mm for penetrating the mold. The alignment airgap dielectric layer was formed by molding an Ecoflex (Ecoflex shore 00-30, Smooth-on Inc., Macungie, PA, USA) elastomer solution inside the mold. The Ecoflex solution was prepared by mixing the base and hardener in a weight ratio of 1:1. After combining the mold and micro-needle, it was soaked in the prepared Ecoflex solution and degassed in a vacuum chamber for 30 min. After hardening in a drying oven at 60 °C for 2 h, the dielectric layer was separated and the dielectric layer of an alignment airgap structure was formed. The fabrication process of the alignment dielectric layer is shown in Appendix A. A flexible sensor was fabricated by integrating the manufactured alignment airgap dielectric layer between film electrodes using a 125-μm-thick polyethylene naphthalate (PEN) coated with a 180 nm thick layer of indium tin oxide (ITO, MTI Korea, Seoul, Korea) as the electrode of the pressure sensor, schematically shown in Figure 1b. The photograph of the prepared pressure sensor of the alignment airgap type is shown in Figure 1c.

Appendix A shows the scheme of the set-up for the capacitive sensor characterization. The shape of the airgap structure of the manufactured dielectric was observed by a digital microscope (AM7115MZTL, Dino-Lite, Taiwan). For the load testing of the prepared pressure sensor, a tension/compression stand (ESM303, Mark-10, USA) and a force gauge having a resolution of 0.5 N (M5-100, Mark-10, USA) were applied. The capacitance of the sensor was evaluated using a semiconductor device analyzer (B1500A, KEYSIGHT, Santa Rosa, CA, USA).

## 3. Results and Discussion

For capacitive pressure sensor, the capacitance depends on the area of electrodes (A), the relative dielectric constant of the material ($\varepsilon_{r}$), and the distance between the top/bottom electrodes (d), based on the capacitance of parallel-plate capacitor:(1)$$C=\frac{A\varepsilon_{r}\varepsilon_{0}}{d}$$
where $\varepsilon_{r}$ is the permittivity of air and $\varepsilon_{0}$ is the permittivity of the dielectric layer. In a sensor based on a porous elastomer dielectric later, closing of the pores with increased applied pressure changes both the geometry and the dielectric constant of the capacitor, resulting in a capacitance change. A high porosity of the dielectric layer can be advantageous for wide detection range and high sensitivity of the sensor.

This work presents a flexible capacitive sensor with high sensitivity and a wide detection range, using a dielectric layer with alignment air gaps. The mechanism of such a pressure sensor is schematically illustrated in Figure 2. The response of the sensor under pressure can be divided into three stages. In the first stage, when a small force is applied to the sensor, the capacitance change depends largely on the contact between the electrodes and the dielectric layer, since the changes of the effective permittivity and the dimensions of the dielectric layer are relatively small. In this stage, the pores and air gaps do not change much, and the response of the device can be attributed to the change of the contact area between the electrodes and Ecoflex dielectric layer. In the second stage, the continuous increase of the pressure starts to squeeze the air gaps and the pores of the dielectric layer. The large thickness change $\Delta d_{2}$ and continuously increased effective permittivity $\Delta \varepsilon_{2}$ result in a significant capacitance change. At this stage, the effect of the dimension change of the dielectric layer may dominate. While the air gaps and pores are squeezed significantly, some air gaps remain open in the dielectric layer. In the third stage, under high pressure, the Ecoflex layer is likely to be densified, and the dimension change is likely to be small since most of the air gaps are closed. At this stage, the capacitance change largely originates from the change of the dielectric constant $\Delta \varepsilon_{3}$ with deformation of the remaining porous structure. Therefore, the capacitive sensor could maintain good sensitivity in a wide range of applied pressure. In addition, during the unloading process, the elastic restoring force of the elastomer allows reopening of the closed air gaps, and the Ecoflex dielectric layer of the sensor returns to the initial state with good reversibility of the sensor response. 

Figure 3 shows a cross-sectional image of the Ecoflex dielectric of the alignment airgap structure. Nine different sensors were manufactured to compare their performance depending on the airgap size and spacing of the pressure sensor. The sizes of the manufactured dielectrics were 300 μm, 400 μm, and 500 μm when the internal airgap spacings were 1 mm, 1.5 mm, and 2 mm, as shown in Figure 3a–i. Uniform airgaps were formed with an error range of ±3.4 μm for the airgap size formed on the cross-section and an error range of ±0.5 μm for the spacing of airgaps. Furthermore, the number of internal airgaps decreased as the size and spacing of airgaps increased. The porosity of the dielectric is an important variable affecting the performance, and the porosity inside the dielectric can be calculated using the size and spacing of the alignment airgaps by the following equation.
(2)$$\mathrm{Porosity}\left(\%\right)=\left(\frac{Pore\;Volume}{Total\;volume}\right)\times 100$$
where pore volume is the volume of the dielectric, excluding the internal airgaps, and total volume is the internal volume of the mold. Table 1 shows the porosity obtained for different sizes and spacing of airgaps. The dielectric based on the fabricated airgap structure showed that the porosity increased as the airgap size increased and the spacing decreased, indicating that the internal porosity increases with the decrease in the spacing between the airgap of the sensor.

The sensitivity (S) of the pressure sensor was calculated by the slope of the relative change of capacitance for the applied external pressure. The sensitivity of the pressure sensor is defined as the ratio between the relative change of capacitance and the change in the applied external pressure [34]:(3)$$S=\delta \left(\frac{\Delta C}{C_{0}}\right)/\delta P$$
where ΔC is the change in the sensor’s capacitance according to the applied pressure, $C_{0}$ is the sensor’s initial capacitance without external pressure, and P is the external pressure applied to the sensor.

Graphs plotted between the relative change of capacitance and the pressure change in the range of 1–100 kPa for nine cases of the fabricated alignment airgap pressure sensor are shown in Figure 4a. After classifying them by their size and spacing of the airgaps inside the dielectric into P1–P9, the change in capacitance with the increase in pressure was measured. As a result, the highest response was observed at P7 (500 μm, 1 mm), and the sensor performance was found to change with the size and spacing of the airgaps, proving that the performance of the sensor increases with the increase in the number of internal airgaps. To compare the pressure detection response and sensitivity of two different structures of the pressure sensors of the alignment airgap structure, the P7 sample that showed the highest response and the bulk Ecoflex without airgap were compared. Figure 4b shows the pressure response comparison graph. The response of the sample that used P7 was approximately 2.95 at 100 kPa, approximately 7 times higher than that of the bulk Ecoflex sample (0.42 at 100 kPa). The pressure point and range were selected according to the pressure measurement requirements across three ranges: low-pressure range (1–500 Pa), middle-pressure range (500–15 kPa), and high-pressure range (15–100 kPa). The sensor fabricated with the alignment airgap structure showed a linear response in every pressure range. The sensitivities of the developed sensor were 1.277 kPa⁻¹ (Figure 4c), 0.045 kPa⁻¹ (Figure 4d), and 0.022 kPa⁻¹ (Figure 4e) in each range, 10.7, 3.75, and 11 times higher than those of the bulk Ecoflex sensor. These results show that the proposed method is promising for improving the sensitivity of pressure sensors. In addition, it was compared with the exiting capacitive pressure sensor based on porous/airgap dielectric shown in Appendix A [34,38,39,40,41].

In addition to sensitivity, the detection limit, response time, stability, and repeatability are important indicators for the performance evaluation of pressure sensors. In this experiment, the P7 sample, which showed the highest performance, was used for sensor evaluation. As shown in Figure 5a, the fabricated sensor can detect a small pressure of 20 Pa. The pressure response curve shows clear steps when pressure is applied or removed and demonstrates an excellent ability to detect external stimuli. Under a quickly applied pressure, the sensor exhibits a quick response and recovery time of approximately 100 ms in the loading and unloading processes (Figure 5b). The sensor’s $\Delta C/C_{0}$ rapidly increases or decreases at the loading/unloading points. Figure 5c shows the results of five repeated inputs of the pressures from low to high pressures (0.5 kPa, 1 kPa, 5 kPa, 40 kPa, and 60 kPa). The rate of change of capacitance showed a tendency to increase in proportion to pressure, while it remained constant under the same pressure. Furthermore, as shown in Figure 5d, stable capacitances were observed when pressures were input step by step (10 kPa, 25 kPa, 50 kPa). Considering the changes in the pressure response curve, the hysteresis of loading and unloading showed a significantly small and negligible capacitance change, as seen in Figure 5e. Furthermore, the dynamic pressure of 50 kPa provided by the tension/pressure tester was measured by the pressure sensor. As shown in Figure 5f, the durability and repeatability of the sensor were tested by repetitive loading/unloading under a pressure of 50 kPa for 10,000 cycles. The stable detection performance of the sensor was demonstrated for several thousands of tests. 

The actual applicability of the proposed sensor to a wearable pressure device was evaluated by monitoring the wrist pulse wave once the proposed sensor was fixed to the wrist, as schematically shown in Figure 6a. The normal pulse and the pulse after exercise on a running machine for 5 min were recorded to verify the usability of the sensor for biometric monitoring. Four sensor responses in five-second intervals are shown in Figure 6b. The fabricated sensor could accurately detect the pulse in real-time; the measured values of the pulse (82 beats/min) and the pulse after exercise (168 beats/min) were normal. As seen in Figure 5, the sensor showed a relative change in the capacitance after exercise for 5 min and the frequency of the pulse signal increased due to the increase in blood pressure. Therefore, it is proved that the proposed sensor can be applied to wearable devices and mobile medical assistive devices.

## 4. Conclusions

This study proposed a new structure of a flexible capacitive pressure sensor of an alignment airgap structure. Unlike the conventional method, the dielectric layer of the alignment airgap structure was fabricated by a simple manufacturing process. The fabricated structure exhibited the highest performance at an airgap size of 500 μm and a spacing of 1 mm. According to the experimental results, the fabricated sensor showed good responses under static and dynamic pressures with high sensitivity, linearity, and repeatability. This sensor showed a high sensitivity of 1.277 kPa⁻¹, a low detection limit of 20 Pa, a fast response speed of 100 ms, excellent stability and recoverability over 10,000 cycles and sensitivity higher by up to 11 times compared to the conventional bulk-Ecoflex pressure sensor. In addition, this sensor successfully detected the wrist pulse waves, demonstrating its applicability to wearable devices.

The pressure sensor of the alignment airgap structure can be applied to E-skin, wearable medical assistive devices, and real-time tactile sensing systems among other applications.

## Figures and Tables

**Figure 1 sensors-22-07390-f001:**
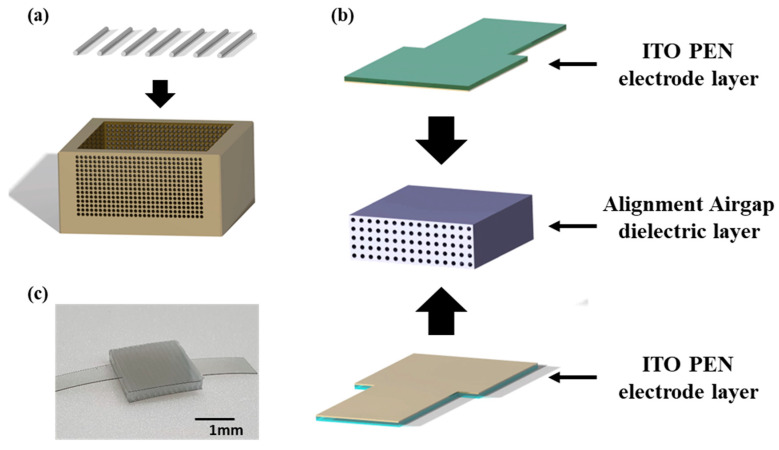
Fabrication process and structure of the device. (**a**) Schematic diagram of the micro-needles and mold for the alignment airgap structure. (**b**) Schematic diagram of the flexible ITO PEN film electrodes (top and bottom) and the alignment airgap dielectric layer (middle). (**c**) Photograph of the capacitive pressure sensor.

**Figure 2 sensors-22-07390-f002:**
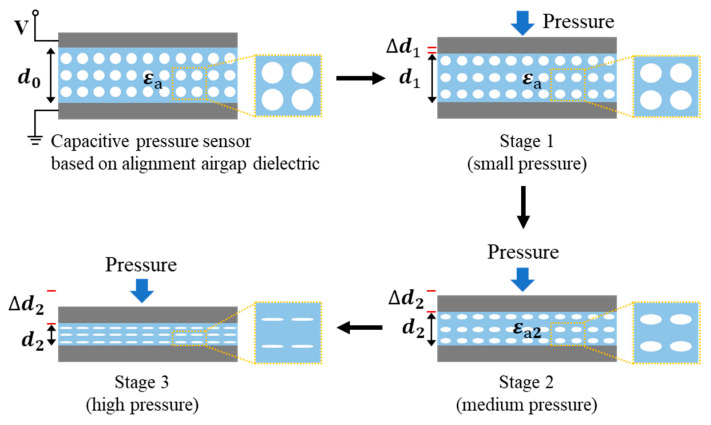
Schematic of the mechanism of the capacitive sensor using an alignment air gap elastomer as a dielectric layer.

**Figure 3 sensors-22-07390-f003:**
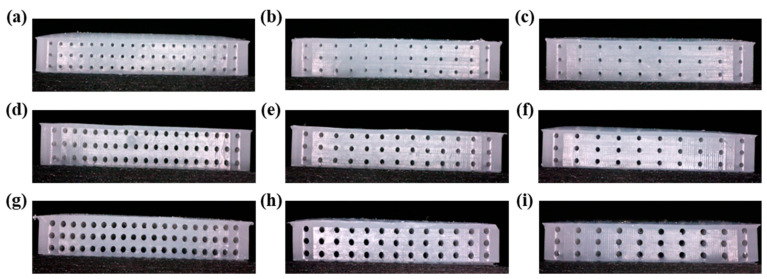
Cross-sectional images of the alignment airgap dielectric layer. (**a**–**c**) pore size: 300 μm, distance: 1 mm, 1.5 mm, 2 mm. (**d**–**f**) pore size: 400 μm, distance: 1 mm, 1.5 mm, 2 mm. (**g**–**i**) pore size 500 μm, distance: 1 mm, 1.5 mm, 2 mm.

**Figure 4 sensors-22-07390-f004:**
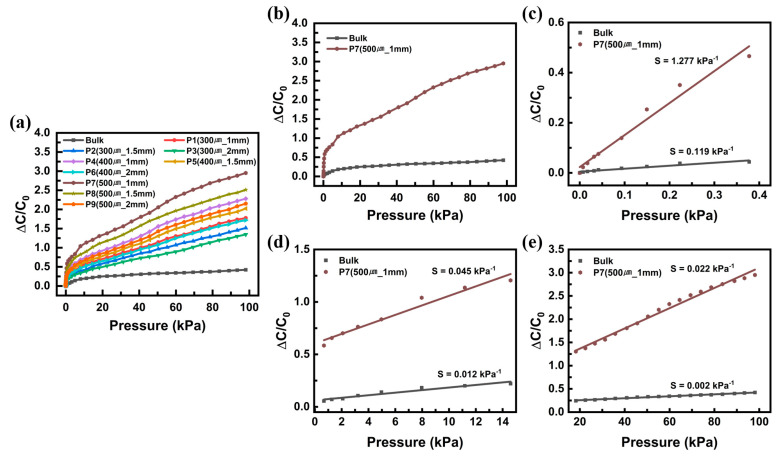
Pressure−response curves of pressure sensors for various structures (**a**) with pore size 300−500 μm (increased by 100 μm), distance 1−2 mm (increased by 0.5 mm). (**b**) Comparison of the pressure sensors’ performance with alignment airgap and with bulk Ecoflex dielectric layer. (**c**–**e**) Variation of the relative difference in capacitance with applied pressure ranges from 1−100 kPa.

**Figure 5 sensors-22-07390-f005:**
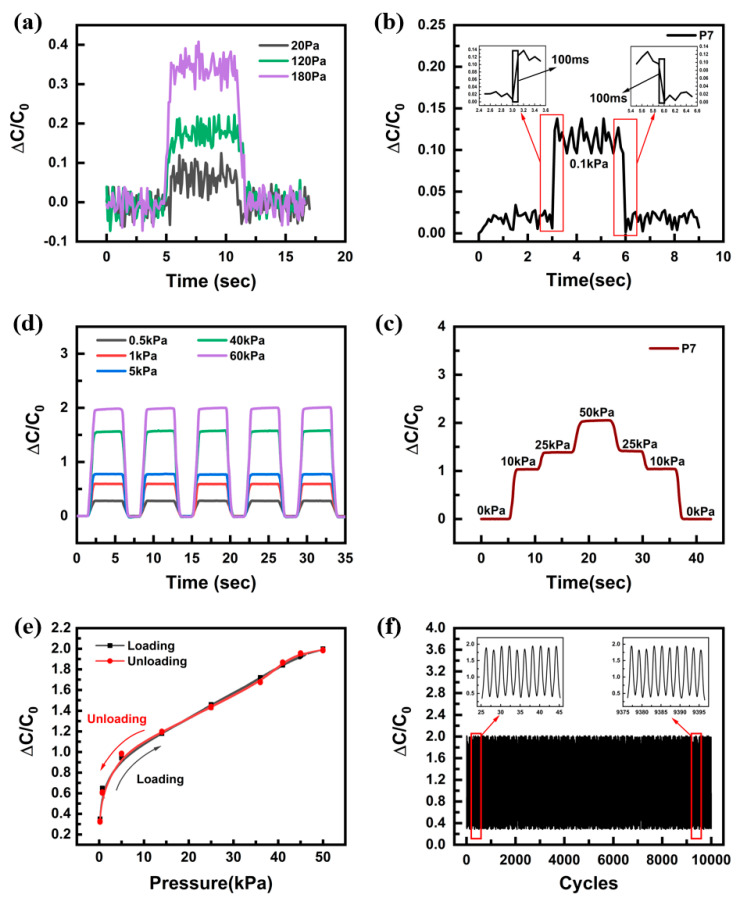
Characterization of the pressure−sensing performance of the alignment airgap pressure sensor. (**a**) Detection limit of the sensor. (**b**) Response and recovery time within 100 ms. (**c**) Squareshaped pressure with different amplitude. (**d**) Capacitance in response to a step increase and decrease in pressure. (**e**) Hysteresis curves of loading/unloading process. (**f**) Capacitance response of pressure sensor during 10,000 loading/unloading cycles at an applied pressure of 50 kPa.

**Figure 6 sensors-22-07390-f006:**
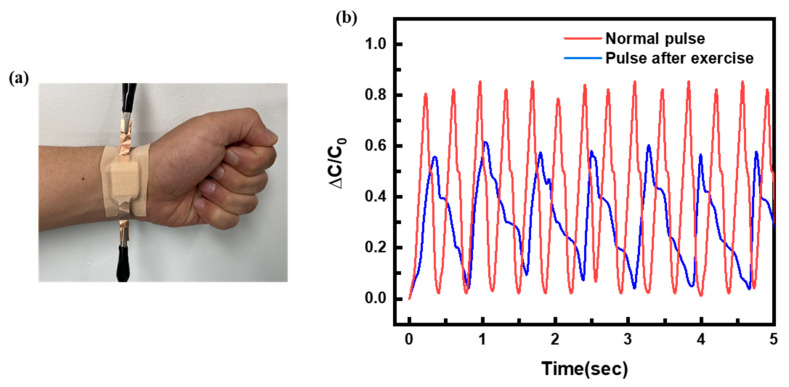
Practical wrist pulse wave applications using a pressure sensor. (**a**) Image of wrist pulse wave measurement of the actual pressure sensor. (**b**) blue indicating the normal pulse and red indicating the pulse after five minutes of exercise.

**Table 1 sensors-22-07390-t001:** Porosity of the dielectric layers for different sizes and spacing.

	1 mm	1.5 mm	2 mm
300 μm	6.715%	4.595%	3.534%
400 μm	11.938%	8.168%	6.283%
500 μm	18.653%	12.763%	9.818%

## Data Availability

Not applicable.

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
