# Peer review of "A Highly Sensitive and Flexible Capacitive Pressure Sensor Based on Alignment Airgap Dielectric"

_sensors, 2022, doi:10.3390/s22197390_

Round 1

Reviewer 1 Report

In this work, the authors propose a flexible capacitive pressure sensors based on a porous Ecoflex dielectric layer with aligned airgaps. The sensor with an alignment airgap structure exhibits excellent a wide working range, high sensitivity, and high operational stability. In addition, the application of the sensor in pulse measurement is demonstrated. Overall, this is an interesting paper on developing flexible pressure sensors with wide range of applied pressures. There are some major and minor comments that need to be addressed before the article is published.

Major comments:

Please specify which properties of the sensor are affected by the irregular airgaps dielectric layer? Which properties of the sensor are improved by the dielectric layer with an alignment airgaps compared to the dielectric layer with an irregular airgaps? Please compare the sensor in this paper with previous sensors in terms of these properties by means of graphs or tables to highlight their advantages.

Minor comments:

1. Please provide the specific dimensions of the sensor in Figure 1c.

2. Please give a physical photo of the sensor test platform.

3. The pressure detection limit of the sensor is 20 Pa, why its measurement range is 1 Pa-100 kPa?

4. Please add the repeatability, error of the sensor.

5. Please improve the quality of Figure 4.

6. The author gives the application test of the sensor at small pressure such as measuring pulse, considering the measurement range of the sensor from 1Pa to 100kPa, please give the application at other pressure range.

Author Response

Point 1: Please specify which properties of the sensor are affected by the irregular airgaps dielectric layer? Which properties of the sensor are improved by the dielectric layer with an alignment airgaps compared to the dielectric layer with an irregular airgaps? Please compare the sensor in this paper with previous sensors in terms of these properties by means of graphs or tables to highlight their advantages.

Response 1: Dielectric material (layer) with irregular air gaps cannot control the distribution of internal air gaps. This results in problems such as unequally distributed air gaps. So, when put under constant pressure, non-uniform pressure distribution might occur.

Also, there are some cons of not being able to achieve the sensor’s reproducibility and repeatability because the air gaps distribution is uncontrollable.

I think the capacitive pressure sensor based on alignment airgap dielectric shown in this work can improve these characteristics.

But comparison experiments with previous sensors were not performed. I was not able to do a comparative experiment. So I added on the mechanism of the capacitive pressure sensor and capacitive pressure sensor that has an air gap in the Results and Discussion section. In addition, put a table that shows the difference between the original capacitive pressure sensor with the airgap by other works (modification range : line 113-144)

Point 2: Please provide the specific dimensions of the sensor in Figure 1c.

Response 2: A scale bar of the Sensor in figure 1c is added.

Point 3: Please give a physical photo of the sensor test platform.

Response 3: The test platform’s production process and measurement setup of the sensor as S1 and S2 of Supplementary data is added.

Point 4: The pressure detection limit of the sensor is 20 Pa, why its measurement range is 1 Pa-100 kPa?

Response 4: There was an error in the notation. I fixed the operating pressure range of Abstract to 20Pa-100kPa. The detection limit of the measured sensor is 20Pa.

The 1pa that is explained in Fig.3 and Fig.4 is an actual measurement range so I did not find the need to change it. (modification range : line 19)

Point 5: Please add the repeatability, error of the sensor.

Response 5: Please consider that not only this work but various research about capacitive pressure sensors present the hysteresis, loading/unloading cycle data shown in Fig.4 c, e, f, etc.

Point 6: Please improve the quality of Figure 4.

Response 6: Improved the quality of Figure 4 will be added on.

Point 7: The author gives the application test of the sensor at small pressure such as measuring pulse, considering the measurement range of the sensor from 1Pa to 100kPa, please give the application at other pressure range.

Response 7: Target of this work was to precede the application of the wearable device at 20Pa-0.5kPa (subtle-pressure), which is the highest sensitivity range from 0-100kPa.

I would need more time to apply in various pressure ranges.

The purpose of this work is to supplement the disadvantages of the previous capacitive pressure sensor with non-uniform distribution of air gaps and to manufacture it in a relatively simple way.

I would like to submit an extended article by applying the sensor structure(thickness, pore control, etc.) and various applications through continuous research.

Reviewer 2 Report

The authors present a pressure sensor that is both sensitive and simple to fabricate. While the work is not groundbreaking, it is sound and worthy of study. Prior to publication, I think the paper can be significantly strengthened with a handful of suggestions:

- It would be worthwhile to include a brief discussion regarding which parameters influence the sensor's sensitivity.

- In addition to modulus, the dielectric constant and geometry can have an impact.

- Perhaps include a simple model predicting your device's capacitance as a function of either pressure or strain. Obviously, the porosity will influence the effective Young's modulus, but note that during compression, the air cavities will collapse, effectively changing your porosity parameter. Because the dielectric constant of Ecoflex is a higher than that of air, this will contribute to your sensitivity. 

- You somewhat arbitrarily pull out three pressure ranges where the sensor locally performs linearly. However, overall, the response is non-linear. Why is the sensitivity so much higher at low pressures? Can the model predict this behavior?

- Line 48 mentions high Young's modulus for capacitive pressure sensors. This is a bit strange, since there are numerous examples of devices and skins that use elastomeric or fluidic dielectrics and/or electrodes. 

- Line 61 states that "In this study, easily soluble particles such as polydimethylsiloxane (PDMS) [36] or Ecoflex [37] are added to the elastomer." The phrase "in this study" is confusing. Which study? Also, soluble particles of PDMS and Ecoflex are added to the elastomer? I am confused as to what this is meant to convey.

- The fabrication section needs a little more detail. After curing the Ecoflex, the needles must be removed, correct? I see no mention of mold release, so I assume that removal was relatively straightforward.

- Your equation for porosity(%) is odd. For pure Ecoflex, the Solid Ecoflex Specified Volume would be equal to the Ecoflex Foam Specified Volume, correct? So then the porosity would be 1? Or 1%? That does not make sense to me. It seems like the equation should be (Pore Volume)/(Total Volume)*100%. 

- In addition to deltaC/C0, I think you should include the raw capacitance values, or at least the C0 values, along with overall dimensions.

- Figure 3 and 4 has some very small text.

- The legend in Figure 3b is slightly different than c, d, and e. This is a minor thing, but I did notice it when glancing at the images.

- Is there an upper limit to the pressure sensing? You tested up to 100kPa, but that was not the failure point.

- You claim a response time of 100ms. What limits this response time? Is it mechanical or electrical? From Figure 4b, it almost looks like you are at the limit of your sampling rate.

- Just from the experimental results, it follows that higher porosity leads to higher sensitivity. In other words, this paper has not found an optimal design. Why not add even larger pores, or pores that are even closer together?

- Do you have thoughts on mass production or more complex geometry?

Author Response

Point 1: It would be worthwhile to include a brief discussion regarding which parameters influence the sensor's sensitivity.

- In addition to modulus, the dielectric constant and geometry can have an impact.

- Perhaps include a simple model predicting your device's capacitance as a function of either pressure or strain. Obviously, the porosity will influence the effective Young's modulus, but note that during compression, the air cavities will collapse, effectively changing your porosity parameter. Because the dielectric constant of Ecoflex is a higher than that of air, this will contribute to your sensitivity.

- You somewhat arbitrarily pull out three pressure ranges where the sensor locally performs linearly. However, overall, the response is non-linear. Why is the sensitivity so much higher at low pressures? Can the model predict this behavior?

Response 1 : I agree with your opinion. So, I added the mechanism for dielectric constant and shape that you suggested in the result and discussion section. Here, I explained the sensing mechanism of the capacitive pressure and also the mechanism of the pressure sensor that uses a dielectric that includes an air gap. ( modification range : line 113 – 142)

Point 2: Line 48 mentions high Young's modulus for capacitive pressure sensors. This is a bit strange, since there are numerous examples of devices and skins that use elastomeric or fluidic dielectrics and/or electrodes.

Response 2 : I think a mansion with a high Young’s modulus can be misunderstood. In order to prevent that, I got rid of the word “the high Young’s modulus” and and modified the contents of various research methods to improve the low sensitivity performance of the existing parallel-plate structure. ( modification range : line 47 – 51)

Point 3: Line 61 states that "In this study, easily soluble particles such as polydimethylsiloxane (PDMS) [36] or Ecoflex [37] are added to the elastomer." The phrase "in this study" is confusing. Which study? Also, soluble particles of PDMS and Ecoflex are added to the elastomer? I am confused as to what this is meant to convey.

Response 3 : Change the word “in this study” to “particle-template method” ( modification range : line 59-60)

Point 4: The fabrication section needs a little more detail. After curing the Ecoflex, the needles must be removed, correct? I see no mention of mold release, so I assume that removal was relatively straightforward.

Response 4 : The contents of the fabrication process have been added to the supplymentary material (Fig. S1)

Point 5: Your equation for porosity(%) is odd. For pure Ecoflex, the Solid Ecoflex Specified Volume would be equal to the Ecoflex Foam Specified Volume, correct? So then the porosity would be 1? Or 1%? That does not make sense to me. It seems like the equation should be (Pore Volume)/(Total Volume)*100%.

Response 5 : I agree with the your opinion. Therefore, the equation was modified to (Pore volume)/(Total Volume)*100% and description of the equation has also been modified. ( modification range : line 156-161)

Point 6: In addition to deltaC/C0, I think you should include the raw capacitance values, or at least the C0 values, along with overall dimensions.

Response 6 : The general way of stating the sensing performance of a capacitive pressure sensor is delta C/C0, which is the relative change in capacitance.

The initial capacitance changes based on each sensor’s thickness, size, and structure. This makes it hard to compare the difference between sensors.

I do aknowledge that the sensor application data is used as the pure capacitance value in various studies. But I also know that stating it as delta C/C0 can also be accepted.

Please consider making the data available.

Point 7: Figure 3 and 4 has some very small text.

Response 7: Changed the text size in Fig 3 and 4

Point 8: The legend in Figure 3b is slightly different than c, d, and e. This is a minor thing, but I did notice it when glancing at the images.

Response 8: Changed legends in Fig c, d and e.

Point 9: You claim a response time of 100ms. What limits this response time? Is it mechanical or electrical? From Figure 4b, it almost looks like you are at the limit of your sampling rate.

Response 9 : Capacitive pressure sensors are known for their fast response/recovery times. Therefore, response/recovery time is an important performance indicator of the sensor in this study. Figure 4b shows performance measurement data for how quickly the capacitance reaches a pressure of 0.1 kPa. 

In the case of the sampling, tension tester (ESM303) and force gauge (mark-10,m5) were used at the maximum speed, which is 1,100mm/min.

Point 10: Just from the experimental results, it follows that higher porosity leads to higher sensitivity. In other words, this paper has not found an optimal design. Why not add even larger pores, or pores that are even closer together?

Response 10: 

The purpose of this work is to supplement the disadvantages of the previous capacitive pressure sensor with non-uniform distribution of air gaps and to manufacture it in a relatively simple way.

And we wanted to check the performance of the sensor for the size and spacing of these alignment airgaps. According to the reviewer's opinion, I would like to conduct research on optimal design (thickness, pore size, distance, etc.) for actual wearable device applications in the future.

The purpose of this work is to supplement the disadvantages of the previous capacitive pressure sensor with non-uniform distribution of air gaps and manufacture it in a relatively simple way. I  also wanted to check the performance of the sensor for the size and spacing of these alignment airgaps.

As your opionion suggests, I would like to conduct a research on opitmal design(thickness, pore size, distance, etc. ) for actual wearable applications in the future.

Point 11: Do you have thoughts on mass production or more complex geometry?

Response 11 : If the research is successful, the simple mold-based manufacturing method presented in this work is expected to be advantageous for mass production applications. And as the research progresses, not only more complex geometries (pore shape, angle, mixed structure), We are considering a study applying various materials (Mxene, Graphene oxide, etc.) that can increase the dielectric constant of a dielectric.

If the research comes out successfully, the simple mold-based manufacturing method presesnted in this work is expected to be advantageous for usage in mass production. And as the research progresses, not only more complex structures (pore shape, angle, mixed structures), but I am also considering a study applying various materials (Mxene, Graphene oxid, etc.) that can increase the dielectric constant of a dielectric.

Round 2

Reviewer 1 Report

The manuscirpt should be accepted as it is.